# COVID-19 Vaccines Currently under Preclinical and Clinical Studies, and Associated Antiviral Immune Response

**DOI:** 10.3390/vaccines8040649

**Published:** 2020-11-03

**Authors:** Swati Jain, Himanshu Batra, Poonam Yadav, Subhash Chand

**Affiliations:** 1Department of Biology, The Catholic University of America, Washington, DC 20064, USA; 80jain@cua.edu (S.J.); 04batra@cua.edu (H.B.); 2CHI Health, Department of Pulmonary Medicine, Creighton University Medical Center, Omaha, NE 68131, USA; poonamyadav.jjm@gmail.com; 3Department of Anesthesiology, University of Nebraska Medical Center, Omaha, NE 68198, USA

**Keywords:** COVID-19, SARS-CoV-2, vaccine development, vaccine platforms, vaccine response, neutralizing antibodies, T-cell responses, convalescent patients

## Abstract

With a death toll of over one million worldwide, the COVID-19 pandemic caused by SARS-CoV-2 has become the most devastating humanitarian catastrophe in recent decades. The fear of acquiring infection and spreading to vulnerable people has severely impacted society’s socio-economic status. To put an end to this growing number of infections and deaths as well as to switch from restricted to everyday living, an effective vaccine is desperately needed. As a result, enormous efforts have been made globally to develop numerous vaccine candidates in a matter of months. Currently, over 30 vaccine candidates are under assessment in clinical trials, with several undergoing preclinical studies. Here, we reviewed the major vaccine candidates based on the specific vaccine platform utilized to develop them. We also discussed the immune responses generated by these candidates in humans and preclinical models to determine vaccine safety, immunogenicity, and efficacy. Finally, immune responses induced in recovered COVID-19 patients and their possible vaccine development implications were also briefly reviewed.

## 1. Introduction

Coronavirus disease 2019 (COVID-19) pandemic has resulted in nearly 50 million infections, claiming more than one million human lives globally (https://coronavirus.jhu.edu). COVID-19 is caused by severe acute respiratory syndrome coronavirus 2 (SARS-CoV-2), a beta-coronavirus that emerged from bats and then transferred to humans through intermediate hosts [1,2,3]. SARS-CoV-2 transmits from human to human via respiratory droplets leading to a respiratory tract infection that can progress to severe pneumonia, multiple organ involvement, and fatal outcomes [4,5,6]. The SARS-CoV-2 infection can be asymptomatic or symptomatic with mild to life-threatening symptoms [6,7,8,9,10]. Irrespective of the severity of symptoms, an infected individual is very likely to spread the infection and especially pose a greater risk to the vulnerable population [11,12,13].

A vaccine is urgently needed to control the current exploding global pandemic of COVID-19 and prevent recurrent epidemics. COVID-19 combined with the seasonal Flu epidemic is expected to further aggravate the situation in terms of diagnosis, co-infections and severity of the disease. An effective vaccine would put a check on the ongoing health or medical crisis and improve the socio-economic status of the society that has severely been impacted due to COVID-19 pandemic [14,15]. Additionally, a return to normalcy with no social distancing or masks can be achieved once all are vaccinated and immune to COVID-19. Hence, eradicating SARS-CoV-2 is very much needed through an effective vaccine, especially in the absence of specific licensed drugs to treat COVID-19.

Since SARS-CoV-2 is closely related to severe acute respiratory syndrome (SARS), Middle East respiratory syndrome (MERS), and other coronaviruses, vaccine design greatly relies on the existing preventive strategies that have been tested for these viruses at the preclinical or clinical level [16,17,18,19]. While virus-neutralizing antibody responses will always be the hallmark for all anti-viral vaccines, some vaccines have also shown the potential to induce protective T-cell responses [16,20,21,22,23]. It is currently not entirely known as to what exactly will prove to be a correlate of protection against this novel coronavirus.

SARS-CoV-2 is an enveloped single-stranded RNA virus. The viral envelope is embedded with spike (S) glycoprotein, the matrix (M) protein, and envelope (E) protein. Encased into this envelope is a positive-sense single-stranded RNA as a viral genome bound to helical nucleocapsid (N) protein (Figure 1) [24,25]. The viral spike protein that mediates entry into the host cell has been identified as one of the preferred vaccine targets. SARS-CoV-2 targets the ACE2 receptor on the host cell via its spike Protein (S), composed of S1 and S2 subunits [24,26]. The S1 subunit contains the receptor-binding domain (RBD) that interacts with the ACE2 receptor, thereby inducing a series of conformational changes facilitating membrane fusion and entry [26]. Due to the critical role played by the spike protein in mediating viral attachment and entry, nearly all SARS-CoV-2 vaccines currently in development are predominantly focused on eliciting protective immune responses targeting the viral spike [21,23,25,26,27,28,29,30]. Although the spike protein is the major component of most vaccines, the technical platform determines how different platforms can modulate the immune responses. Therefore, safety, immunogenicity, and efficacy will majorly depend on the vaccine approach or delivery platforms. In general, vaccine platforms are broadly categorized into six types: live attenuated viral vaccine, inactivated virus vaccine, recombinant viral-vectored vaccines, protein subunit vaccines, virus-like particles (VLPs), and nucleic acid-based (DNA or mRNA) vaccines (https://www.vaccines.gov/basics/types). The relative progress of the vaccine candidates towards different stages of clinical development is shown in Figure 2.

Here, we will discuss the predicted and observed immune responses for major vaccine candidates depicting each platform that are currently in preclinical or clinical phases of development to prevent the SARS-CoV-2 infection. Furthermore, we discussed the immune responses elicited in recovered COVID-19 patients that can provide useful vaccine design insights.

## 2. Major COVID-19 Vaccine Candidates and Their Responses

Vaccine safety and efficacy vary for a protein/DNA/RNA vaccine or with the type of adjuvant/vector used in the vaccine formulation, and even with the route of administration. Similarly, whether the SARS-CoV-2 spike is made to express endogenously in the vaccines as part of the nucleic acid (DNA/mRNA) vaccine approach or administered as a recombinant protein antigen for immunization can induce considerable variations in vaccine responses that can subsequently influence the vaccine efficacy. Different types of vaccine platforms currently in trials for COVID-19 are shown in Figure 3. While the vaccine efficacy is majorly assessed through adaptive immunity components, such as the induction of robust virus-neutralizing antibody responses, the innate arm of the immune defense plays a critical role in resulting in effective adaptive responses [31,32]. The adjuvants present in the vaccine formulation primarily activate innate responses and enhance the adaptive immune responses governing the effectiveness of the vaccine [33]. Mechanistically, adjuvants act as ligands for TLRs (Toll-like receptors) or PRRs (pattern recognition receptors), and the specific interaction of each adjuvant with the respective receptors determines the outcome of response [33]. A well-known adjuvant, alum, for instance, is known to activate Nalp3 inflammasome and activate Th2 T-cell response [34,35]. The adjuvant effect can also be seen for some vaccines that lack any added adjuvants. Killed or inactivated vaccines, live-attenuated vaccines, and viral-vectored vaccines contain their own PAMPs (pathogen associated molecular patterns) that can serve as built-in adjuvants [36]. Discussed below are the major COVID-19 vaccine candidates under different vaccine technology platforms with their composition, respective responses, and development stage, along with the information summarized in Table 1.

### 2.1. Inactivated/Live Attenuated Virus Vaccine

Vaccine development using the weakened or inactivated virus has been a traditional approach for decades, including measles and the first iteration of the Salk and Sabin polio vaccines [37,38]. A live attenuated virus vaccine is weakened either by serial passaging in animal/human cells or by altering the viral genetic code to dampen the virus [39,40]. A virus can also be inactivated using chemicals such as formaldehyde and heat [41,42]. Codagenix (New York) partnered with Serum Institute of India, and Indian immunologicals Ltd. and Griffith University, and Mehmet Ali Aydinlar University have generated a genetically altered weakened form of SARS-CoV-2 as a live attenuated vaccine, currently under preclinical evaluation (draft landscape of COVID-19 candidate vaccines). Live virus vaccines are usually highly immunogenic; therefore, one dose is often enough to produce a substantial immune response; however, there is always a high risk of the reversion of attenuated strain to a pathogenic one.

Sinovac Biotech in Beijing, China, has been testing an inactivated form of SARS-CoV-2 in preclinical studies [43]. Sinovac’s SARS-CoV-2 virus vaccine candidate (PiCoVacc) has been produced in Vero cells and inactivated using β-propiolactone. This vaccine was tested in mice, rats, and non-human primates (NHPs) in two doses (3 and 6 µg). PiCoVacc reportedly induced antibodies in these preclinical animal models capable of neutralizing 10 representative strains of SARS-CoV-2. Furthermore, partial-to-complete protection was observed in macaques after three immunizations with PiCoVacc against the SARS-CoV-2 challenge [43]. Notably, the histopathological assessment showed no pathological changes in vaccinated macaques’ vital organs [43]. Furthermore, in contrast to a live virus, the inactivated virus vaccine did not result in a percent change in lymphocytes (CD3+, CD4+, or CD8+) or a cytokine storm that is a leading cause of death in SARS-CoV-2 infected individuals [5,6,8,43]. Another inactivated vaccine candidate, BBIBP-CorV, developed by Sinopharm, China, was also assessed for efficacy in multiple animal models, including NHPs [44]. This vaccine elicited robust neutralizing antibody titers even with the lowest dose (2 μg) tested. Additionally, BBIBP-CorV was able to confer protection in macaques without any antibody-dependent enhancement [45]. Both Sinovac (NCT04582344) and Sinopharm (NCT04560881) vaccine candidates are currently under Phase 3 study. Although it is encouraging that these inactivated vaccines could elicit desirable humoral responses in preclinical models (draft landscape of COVID-19 candidate vaccines), the complete inactivation of the virus would be critical for the safety of the vaccines in humans.

### 2.2. Viral Vectored Vaccine

Recombinant viral vectored vaccines are among the most common candidates leading in the race of SARS-CoV-2 vaccines, with four of them in the clinical phase and several in the preclinical development stages. Adenovirus type-5 (Ad5), Ad26, and vesicular stomatitis virus (VSV) are commonly used viral vectors for this vaccine platform. Developed by CanSino Biologics, China, a recombinant Ad5 vectored COVID-19 vaccine expressing the spike glycoprotein of a SARS-CoV-2 has been assessed recently in Phase 1 non-randomized study for safety and immunogenicity (NCT04568811). A total of 108 healthy participants were enrolled in Wuhan, China, in the age group of 18–60 years for this trial [21]. Due to adenovirus-based vaccines’ undesirable immunogenicity risks, the vaccine was tested in low, medium and high doses. All vaccine recipients had induced anti-RBD antibodies irrespective of the dose after 14 days, that peaked 28 days post-vaccination. Though not examined, antibodies against epitopes on the spike other than RBD are also expected to be induced through this vaccine approach. These responses would also govern the extent of the vaccine’s success in the subsequent phases of the clinical trials. Additionally, the Ad5-vectored vaccine generated antibodies also demonstrated in vitro neutralization of the SARS-CoV-2 virus [21].

Additionally, T-cell responses in the form of the release of IFNγ, TNFα, and IL-2 were detected from CD4+ and CD8+ T cells that peaked at day 14 post-vaccination in all dose groups [21]. However, both humoral and cell-mediated responses were significantly higher in the high dose group compared to the middle and low dose groups. Notably, antigen-specific antibodies and T-cell responses were partially reduced in the recipients with pre-existing immunity against adenovirus [21]. Most of the participants also suffered from mild to moderate adverse reactions, such as pain at the injection site, fever, fatigue, headache, and muscle pain post-vaccination. However, no severe adverse reactions were reported at least until 28 days post-vaccination [21]. Since the responses were observed only for 28 days, a follow-up study would be required to evaluate the immune response’s durability. Ad26-vectored COVID-19 vaccine is another candidate developed by Johnson & Johnson that demonstrated protection in NHPs with the advantage of being less immunogenic than Ad5 [46]. ChAdOx1-nCoV-19 vaccine vectored with chimpanzee adenovirus has also been developed by the University of Oxford and AstraZeneca [47,48]. The same platform was employed for MERS and tuberculosis (TB) with promising results in human clinical trials [17,49]. Unlike Ad5/Ad26 vectors, ChAdOx1 has far less pre-existing immunity in humans, which is a critical determinant of this platform’s efficacy. ChAdOx1-nCoV-19, indeed, has shown a robust induction of neutralizing antibody and T-cell responses, in conjunction with a reduction in viral titers in rhesus macaques. Importantly, the Phase 1/2 study has shown ChAdOx1-nCoV-19 to be safe and also effective in producing cellular and humoral responses [17,48,50]. Ad5-vectored (NCT04540419), Ad26-vectored (NCT04505722) and ChAdOx1-nCoV-19 (NCT04540393) vaccine are all currently in Phase 3 clinical trials. Other vaccines based on VSV and modified vaccinia virus Ankara (MVA) viral vectors have also shown promising results in preclinical animal models (draft landscape of COVID-19 candidate vaccines).

### 2.3. mRNA Vaccine

Like the DNA vaccine, no mRNA vaccine has been approved for human use. However, preclinical studies conducted for mRNA-based influenza and zika virus vaccine have demonstrated the induction of protective responses [51,52]. An mRNA vaccine, mRNA-1273 for SARS-CoV-2 encoding a prefusion stabilized form of its Spike (S) protein, has been co-developed by researchers at the National Institute of Allergy and Infectious Diseases (NIAID) and at Moderna (Cambridge, MA) [28,53]. This is the first mRNA vaccine to go into clinical trials for the safety and immunogenicity assessment. The mRNA vaccine concept is supported by the principle that SARS-CoV-2 itself is a (+) ss-RNA virus. In order to block initial virus interactions and spike mediated viral entry into the host cell, spike-specific mRNA was utilized as a vaccine target and delivered by encapsulating into lipid nanoparticles (LNPs). As a general principle, the mRNA vaccine upon delivery is expected to enter cells and translate or encode the target protein in the cell cytoplasm [53]. After translation, this foreign protein is released from the cells and encountered by APCs, and results in processing and major histocompatibility complex I (MHC I) subsequent MHC II-based presentation of the target protein. This cascade of events leads to the engagement and activation of B-cells and T-cells to orchestrate both humoral and cell-mediated antigen-specific responses. While MHC I presentation causes the activation of antigen-specific CD8+ T cells, MHC II presentation facilitates CD4+ T cell and B-cell activation followed by mounting of an antibody response [53,54,55].

Additionally, macrophages’ uptake of secreted target antigen results in the secretion of pro-inflammatory cytokines and chemokines, activating the innate arm of immune defense [53]. Results from the Phase 1 study for this novel vaccine unveiled that mRNA-based vaccines can safely induce binding and virus-neutralizing antibodies against the spike protein in all the vaccine recipients after two doses. Th1-based CD4+ T-cell responses, and to a lesser extent, CD8+ T-cell responses, were also observed in vaccine recipients. No severe side effects of the vaccine were reported [28]. Though efficacy evaluation and correlates of protection are not currently known, preclinical studies performed to evaluate mRNA-1273 vaccine responses in mice demonstrated the induction of neutralizing responses post-vaccination and even protection after challenge with SARS-CoV-2 [56]. The safety and immunogenicity data have recently been published for the phase 1 trial of mRNA-1273 in both young and older adults, assessing a dose range from 25 to 100 μg (NCT04405076). A large phase 3 efficacy trial (NCT04470427) evaluating a 100 μg dose has already begun to further assess the mRNA-1273 vaccine in approximately 30,000 adult volunteers [28,56]. Alternatively, Pfizer and BioNTech vaccine candidates based on mRNA encoding SARS-CoV-2 RBD complexed with lipid nanoparticles are also under Phase 3 study (NCT04368728). Other mRNA candidates developed by CureVac (NCT04515147) and Arcturus/Duke-NUS (National University of Singapore) (NCT04480957) are also rapidly progressing through Phase 2 trials for assessment.

### 2.4. DNA Vaccine

Although there are no approved licensed DNA vaccines for humans, many DNA vaccine candidates are in preclinical and even clinical trials [57,58]. Once expressed, the protective protein antigen can be processed endogenously and presented by an antigen-presenting cell (APC) complexed with MHC. DNA delivered via different viral and non-viral vaccine platforms enter the cell via endocytosis and trigger an innate immune response through innate immune-system receptors, such as Toll-Like Receptor 9 (TLR9) present in endosomes [59,60]. MERS-CoV vaccine (INO-4700) and zika vaccine candidate (GLS-5700) are the DNA vaccines that are currently in clinical testing [19,61,62]. In recipients of INO-4700, durable neutralizing antibodies (nAbs) and T cell immune responses were observed with a seroconversion rate of 96% [61]. SARS-CoV-2 spike protein-coding DNA vaccine, INO-4800, has recently been developed and evaluated for immunogenicity in mice and guinea pigs. In this preclinical testing, INO-4800 induced immunoglobulin G (IgG) responses against spike protein just after a single dose. Additionally, virus-neutralizing antibodies were observed in these immunized animals with a demonstrated potential to compete with ACE2 binding to the SARS-CoV-2 Spike protein [63]. INO-4800 (NCT04336410), along with three other DNA vaccines, are currently under Phase 1/2 study with several in the preclinical studies (draft landscape of COVID-19 candidate vaccines).

By virtue of highly flexible and cost-effective features of the DNA vaccine platform, a series of prototypic DNA vaccine candidates based on differing lengths of the SARS-CoV-2 spike encoding gene have also been evaluated for immunogenicity and efficacy in rhesus macaques via the intramuscular route without an adjuvant [23]. These candidates included six variants of the SARS-CoV-2 S protein based on the site of truncation; full-length (S), cytoplasmic tail deletion mutant (S.dCT), transmembrane domain and cytoplasmic tail deletion mutant (S.dTM), S1 subunit (S1), receptor-binding domain (RBD), and a prefusion stabilized ectodomain with two proline mutations (S.dTM.PP). Spike-specific binding and virus-neutralizing antibody responses exhibiting various subclasses and effector functions were observed after boost immunization. Higher antibody-dependent complement deposition (ADCD) responses were also observed in the S and S.dCT groups, whereas higher natural killer (NK) cell activation was observed in the RBD and S.dTM.PP groups. Cellular immune responses targeted to a pool of S peptides were also detected in the majority of the vaccinated macaques shown by IFN-γ enzyme-linked immune absorbent spot (ELISPOT) assays after the booster dose. Intracellular cytokine staining assays showed an induction of S-specific IFN-γ+ CD4+ and CD8+ T cell responses, with relatively reduced responses observed in the shorter S1 and RBD immunogen groups. Moderate S-specific IL-4+ CD4+ and CD8+ T cell responses were observed, indicating a bias towards Th1 over Th2 cellular immune responses. The vaccinated animals challenged with SARS-CoV-2 showed a significant reduction of viral RNA, demonstrating the protective efficacy. However, a minimal level of protection was seen in the S.dTM group highlighting the importance of prefusion ectodomain stabilization for an effective vaccine [23]. More optimal protection was collectively achieved with the full-length S immunogen than soluble S immunogens and smaller fragments.

While the DNA vaccine platform might have the potential to generate protective responses, the risk of random insertion mutagenesis resulting from integration in the host genome, anti-DNA antibody generation, and auto-immune diseases will remain [64].

### 2.5. Recombinant Protein-Based Vaccine

Protein components of the targeted pathogen that can stimulate protective responses are considered useful for the subunit vaccines [65,66]. Typically, subunit vaccines constitute surface or structural proteins. For viral vaccine candidates, the spike protein required for virus attachment and entry into the host cell is often targeted with a rationale to elicit responses that can block viral entry [67,68,69]. SARS-CoV-2 spike (S) protein is composed of the S1 subunit that encompasses- N-Terminal Domain (NTD), RBD, and C-Terminal Domain (CTD), and S2 subunit that contains fusion peptide, the transmembrane domain, and the cytoplasmic tail [24,70]. Spike protein S assembles as a homotrimer and is heavily glycosylated [24,70]. Variants of SARS-CoV-2 spike antigens have been analyzed in a preclinical study in rabbits by Ravichandran et al. [71]. In this study, rabbits were immunized with spike-ectodomain (S1 + S2), S1 domain, receptor-binding domain (RBD), and S2 domain. They observed that all but the S2 group could induce strong nAb responses. Higher titers of high-affinity nAbs were observed in the RBD immunized group, supporting it as a promising vaccine candidate [71]. Similar RBD-induced potent responses were also observed in the preclinical studies of SARS, MERS, and other coronaviruses [72,73]. Both full-length S-trimer and RBD-based vaccine candidates such as that developed by Novavax (2020-004123-16) has entered Phase 3 study, followed by candidates developed by Sanofi Pasteur (NCT04537208), Clover Biopharmaceuticals/GSK (GlaxoSmithKline plc) (NCT04405908), Vaxine Pty Ltd. (NCT04453852), etc. in Phase 1/2 and Phase 1 trials, especially with different adjuvants (draft landscape of COVID-19 candidate vaccines).

Additionally, virus-like nanoparticles (VLP) mimicking the viral structural features but devoid of the genome are also planned to be evaluated in clinical trials. Medicago (NCT04450004) SARS-CoV-2 VLP (CoVLPs) vaccine adjuvanted with GSK proprietary adjuvant system is under Phase 1 study (draft landscape of COVID-19 candidate vaccines) [74]. Similarly, many other VLP-based vaccines are in preclinical trials. A non-invasive oral vaccine for SARS-CoV-2 designed by Vaxart, aimed at eliciting mucosal immune responses, is an additional novel vaccine candidate in the pipeline (https://investors.vaxart.com/news-releases/news-release-details/vaxart-announces-fda-clearance-ind-application-oral-covid-19). Adjuvants are commonly used in protein-based vaccine formulations that might govern the outcome of responses as well as the protective efficacy of a vaccine.

Exploring the intra-cutaneous route of administration for COVID-19 vaccine, a minimally-invasive microneedle array (MNA) vaccine delivery platform is also under development. Since the skin is abundantly rich with immune cells, specifically Langerhans cells, it is a robust target for immunization to generate potent immune responses [75,76]. SARS-CoV-2 spike protein fused to a trimerization motif-foldon (derived from phage T4 fibritin) was embedded into MNA for the SARS-CoV-2 vaccine [27]. A similar strategy was also applied previously for the MERS vaccine using its spike protein [27]. Furthermore, virus-specific nAbs were detected for the MERS-MNA vaccine in mice, while the neutralizing responses are yet to be determined for the SARS-CoV-2-MNA vaccine [27]. Though MNA-mediated immunization demonstrated potent adaptive responses based on the preclinical study in mice, the actual efficacy and protection will be obtained from future human clinical trials. By targeting the skin microenvironment through microneedle array, this platform utilizes physical adjuvant to generate antigen-specific responses with relatively low doses. Further studies to determine the potency of adaptive/innate responses in SARS-CoV-2-MNA vaccine recipients in clinical stages would be highly sought.

## 3. Insights from Immune Responses Elicited in the Recovered Patients for Vaccine Development

A more in-depth understanding of the immune responses that are elicited in recovering COVID-19 patients might provide useful insights into vaccine design. As a hallmark for an effective vaccine, the induction of virus-neutralizing antibody responses is often considered inevitable. With the recent approval of convalescent plasma therapy, the U.S. FDA ignited great interest in its therapeutic potential. However, several independent studies performed at multiple locations globally have shown that nAbs titers in the plasma of mildly symptomatic patients recovering from COVID-19 are highly variable [77,78,79]. A similar study, based on 68 convalescent SARS-CoV-2 patients, by Robbiani et al. showed that on average, the nAb titers remained low in these patients, even undetectable in 18% of them, while only 3% had high nAb titers [80]. Variability in nAbs titers was also reported by Wu et al., based on a cohort study of 175 convalescent COVID-19 patients [81]. Additionally, they observed higher nAb titers in older rather than younger people. Both studies also confirmed the presence of spike and RBD-specific antibodies, titers of which directly correlated with virus neutralization. Interestingly, RBD-specific antibodies were found to be effective even at much lower titers when tested for virus neutralization in in vitro assays.

Furthermore, RBD-specific B-cell precursors were identified to be commonly prevalent in patients based on antibody sequencing data [80,82,83]. Studies have also shown that several epitopes were targeted exclusively in the RBD region for antibody generation in natural infections, and the majority of such antibodies proved potent in virus neutralization [80]. Other than spike-directed responses, antibodies targeting the nucleoprotein (NP) of SARS-CoV-2 were also observed with the potential of virus neutralization in the COVID-19 infected patients [84]. However, expected variability in immune responses is due to many factors such as age, sex, geographical location, and prevalent strain of the virus, as reviewed above. Considering wide variations in nAbs titers in the convalescent patients and a lack of correlation with the disease and the recovery’s mild outcome, it is difficult to say if vaccine-elicited nAbs would be enough for adequate protection against SARS-CoV-2.

In order to mount a robust immune response against an invading pathogen, both adaptive and innate arms of the immune system work in conjunction. Though antibodies are traditionally considered as necessary molecules of immune defense, their generation relies on effective cross-talk with the T-cells. Griffoni et al. detected SARS-CoV-2-specific CD4+ and CD8+ T-cells in convalescent patients based on predicted T-cell epitopes spanning the whole viral genome. Spike-specific CD4+ T-cell responses were exceptionally prevalent in all infected individuals and were notably correlated with the anti-spike RBD antibody responses [85]. Unlike SARS infections where T-cell responses were predominantly directed to the viral spike, in SARS-CoV-2 infections, M and N proteins along with a spike were targeted to elicit T-cell responses.

Furthermore, CD4+ T-cells responses were also directed towards the non-structural antigens such as; nsp4, ORF3s, ORF7a, nsp12, and ORF8 [85]. This suggests that although spike/RBD is a prime vaccine target for all current vaccine development approaches, the inclusion of other structural and non-structural viral antigens might better recapitulate a scenario occurring in the convalescent patients after natural infection. Overall, T-cell immunity has been positively correlated with improved recovery in infected patients, and SARS-CoV-2 infected individuals with severe disease have been shown to undergo T-cell lymphopenia [86]. Some studies have also predicted the occurrence of T-cell exhaustion in COVID-19 patients [87]. Although virus targeted T-cell immune responses might be beneficial to consider for vaccine development, T-cell immunopathologies should be monitored in vaccine recipients, as these undesirable responses have previously been observed for SARS vaccine candidates.

Developing a vaccine that can cross-protect against similar coronaviruses would be an ideal consideration for the future. SARS shares about 80% of the sequence homology with SARS-CoV-2 at the genomic level, with both viruses utilizing the ACE2 receptor for the host cell entry [24,88,89]. While these commonalities have shown cross-reactive responses as observed by many studies, cross-protection could not become evident. This cross-reactivity is majorly attributed to the conserved viral antigenic epitopes and would be worth considering while designing a broadly cross-protective vaccine against related coronaviruses. However, such attempts should be made cautiously as the presence of cross-reactive antibodies has also been previously observed to enhance the infection through antibody-dependent enhancement (ADE) in case of other viral infections, including SARS [18,90]. Thus, despite the urgency of a SARS-CoV-2 vaccine, both the safety and efficacy should be critically evaluated before licensing a vaccine.

## 4. Rapid Nature of the Vaccine Development and Its Drawbacks

Owing to the advancements in vaccine development in recent decades, the time-frame for bringing a vaccine from bench to bedside has considerably shortened. Ebola and zika vaccine development exemplify this rapid clinical translation [91]. While it is very much possible to develop any vaccine with a targeted approach in months to years’ time window, rigorous evaluation in large scale human studies with extended follow-up studies are essential to determine the durability of responses and long-term vaccine efficacy. Additionally, many vaccine studies enroll young, healthy volunteers to assess vaccine efficacy and rapidly progress through vaccine development stages. Due to the urgent need for the COVID-19 vaccine, if a vaccine is licensed based on healthy people’s safety and efficacy, then the response of high-risk people (elderly individuals, children, pregnant women, nursing mothers, etc.) to the vaccine will remain unknown. COVID-19-related immunopathologies observed in severe cases of SARS-CoV-2 infected individuals pose the greatest risk for the safety of a vaccine [6,8]. Further determination of risk due to interaction between the vaccine and virus-induced responses after natural infection in vaccinated individuals will remain a critical focus.

## 5. Conclusions

We are in the very initial stages of understanding the interaction between the immune system and SARS-CoV-2 vaccines to mediate protection and/or susceptibility to COVID-19. However, basic viral immunology knowledge can serve to design a vaccine. Currently, hundreds of COVID-19 vaccine candidates are in development, and success is unknown. However, owing to the concerted efforts made around the globe in a short period to end this pandemic, the likelihood of finding a successful candidate/s is relatively high, especially with the use of a variety of vaccine platforms. Hopefully, the critical evaluation of vaccine candidates for their safety, efficacy, long-term immunity, and protection in widespread population groups will soon bring the COVID-19 pandemic to an end.

## Figures and Tables

**Figure 1 vaccines-08-00649-f001:**
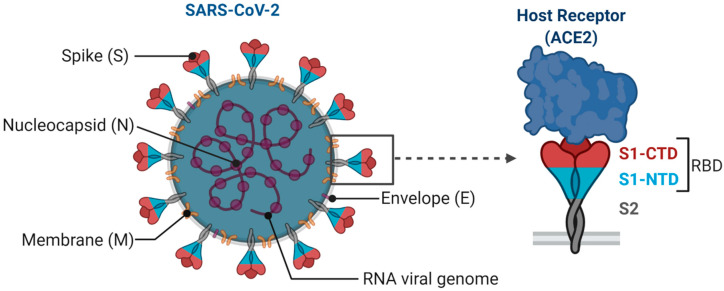
Structural features of SARS-CoV-2. Spike (S) glycoprotein, the membrane (M) protein, and envelope (E) protein are embedded in the viral envelope. The RNA genome is complexed with the nucleocapsid (N) protein. Virus spike trimer is enlarged to depict its key subunits (S1 and S2) and N-terminal domains (NTD) and C-terminal domains (CTD) in the S1 subunit encompassing receptor-binding domain (RBD). S protein targets the host cell receptor, ACE2, through RBD in S1 subunit (Created with Biorender.com).

**Figure 2 vaccines-08-00649-f002:**
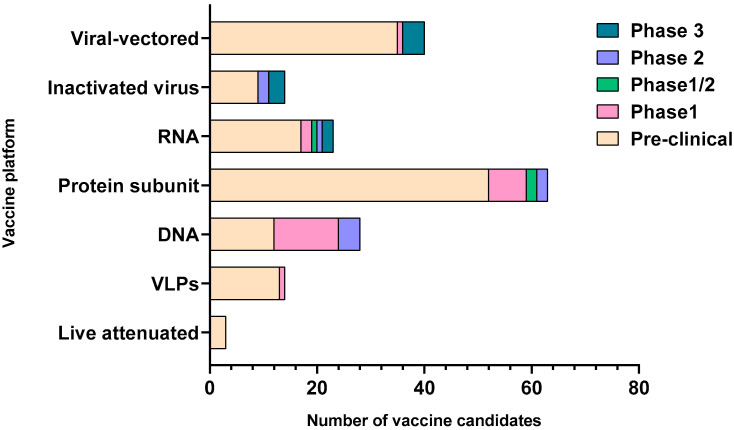
Graphical representation of the vaccine candidates with respect to their clinical stages of development. The number of candidates on the *x* axis are compared for each of the major vaccine platforms shown on the *y* axis. The clinical stage for the vaccine candidates in each platform is depicted by color-coded legends on the right of the graph. The graph is constructed based on the data obtained from the World Health Organization; draft landscape of COVID-19 candidate vaccines and Coronavirus Vaccine tracker. VLPs-virus-like particles, RNA-ribonucleic acid, DNA-deoxyribonucleic acid.

**Figure 3 vaccines-08-00649-f003:**
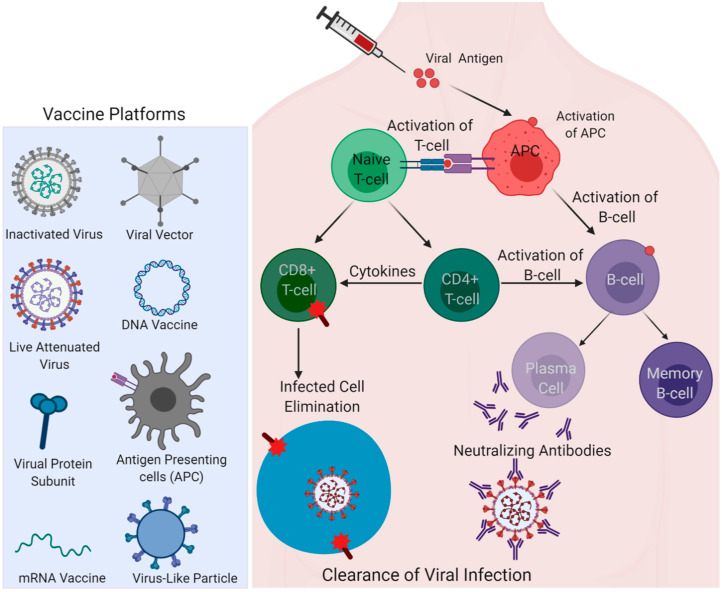
Different vaccine platforms for the COVID-19 vaccine. Left; viral and non-viral vaccine delivery platforms are shown. Right; antigen induced activation of antigen-presenting cells (APCs); a nexus of innate and adaptive immune arms activating T-cell and B-cell immunity by fundamental immunological pathways are represented to show how vaccine-elicited immune responses lead to the clearance of infection. (Created with Biorender.com).

**Table 1 vaccines-08-00649-t001:** Various types of vaccines, their composition, developer, and stage of development.

Types of Vaccines	Name of Vaccine/Developer	Composition	Stage of Development
Inactivated Virus	Sinovac	Inactivated Virus	Phase 3
	Wuhan Institute of Biological Products/Sinopharm		Phase 3
	Beijing Institute of Biological Products/Sinopharm		Phase 3
	Institute of Medical Biology, Chinese Academy of Medical Sciences		Phase 1/2
	Research Institute for Biological Safety Problems, Rep. of Kazakhstan		Phase 1/2
	Beijing Minhai Biotechnology Co., Ltd.		Phase 1
	Bharat Biotech		Phase 1/2
Viral vectored	University of Oxford/AstraZeneca	ChAdOx1-S	Phase 3
	CanSino Biological Inc./Beijing Institute of Biotechnology	Adenovirus Type 5 Vector	Phase 3
	Gamaleya Research Institute	Adeno-based (rAd26-S + rAd5-S)	Phase 3
	Janssen Pharmaceutical Companies	Ad26COVS1	Phase 3
	ReiThera/LEUKOCARE/Univercells	Replication defective Simian Adenovirus (GRAd) encoding S	Phase 1
	Institute of Biotechnology, Academy of Military Medical Sciences, PLA of China	Ad5-nCoV	Phase 1
	Vaxart	Ad5 adjuvanted Oral Vaccine platform	Phase 1
	Ludwig-Maximilians—University of Munich	MVA-SARS-2-S	Phase 1
	Institute Pasteur/Themis/Univ. of Pittsburg Center for Vaccine Research (CVR)/Merck Sharp & Dohme	Measles-vector based	Phase 1
	Beijing Wantai Biological Pharmacy/Xiamen University	Intranasal flu-based-RBD	Phase 1
RNA	Moderna/NIAID	LNP-encapsulated mRNA	Phase 3
	BioNTech/Fosun Pharma/Pfizer	3 LNP-mRNAs	Phase 3
	Curevac	mRNA	Phase 2
	Arcturus/Duke-NUS	mRNA	Phase 1/2
	Imperial College London	LNP-nCoVsaRNA	Phase 1
	People’s Liberation Army (PLA) Academy of Military Sciences/Walvax Biotech.	mRNA	Phase 1
DNA	Inovio Pharmaceuticals/International Vaccine Institute	DNA plasmid vaccine with electroporation	Phase 1/2
	Osaka University/AnGes/Takara Bio	DNA plasmid vaccine + Adjuvant	Phase 1/2
	Cadila Healthcare Limited	DNA plasmid vaccine	Phase 1/2
	Genexine Consortium	DNA Vaccine (GX-19)	Phase 1/2
Protein Subunit	Novavax	Full length recombinant SARS CoV-2 glycoprotein nanoparticle vaccine adjuvanted with Matrix M	Phase 3
	Anhui Zhifei Longcom Biopharmaceutical/Institute of Microbiology, Chinese Academy of Sciences	Adjuvanted recombinant protein (RBD-Dimer)	Phase 2
	Kentucky Bioprocessing, Inc	RBD-based	Phase 1/2
	Sanofi Pasteur/GSK	S protein (baculovirus production)	Phase 1/2
	Clover Biopharmaceuticals Inc./GSK/Dynavax	Native-like Trimeric subunit Spike Protein vaccine	Phase 1
	Vaxine Pty Ltd./Medytox	Recombinant spike protein with Advax™ adjuvant	Phase 1
	University of Queensland/CSL/Seqirus	Molecular clamp stabilized Spike protein with MF59 adjuvant	Phase 1
	Medigen Vaccine Biologics Corporation/NIAID/Dynavax	S-2P protein + CpG 1018	Phase 1
	Instituto Finlay de Vacunas, Cuba	RBD + Adjuvant	Phase 1
	FBRI SRC VB VECTOR, Rospotrebnadzor, Koltsovo	Peptide	Phase 1
	West China Hospital, Sichuan	RBD (baculovirus production expressed in Sf9 cells)	Phase 1
	University Hospital Tuebingen	SARS-CoV-2 HLA-DR peptides	Phase 1
	COVAXX	S1-RBD-protein	Phase 1
VLP	SpyBiotech/Serum Institute of India	RBD-HBsAg VLPs	Phase 1/2
	Medicago Inc.	Plant-derived VLP adjuvanted with GSK or Dynavax adjs.	Phase 1

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
