# Peer review of "COVID-19 Vaccines Currently under Preclinical and Clinical Studies, and Associated Antiviral Immune Response"

_vaccines, 2020, doi:10.3390/vaccines8040649_

Round 1

Reviewer 1 Report

Dear Authors,

your manuscript need some revision to improve it and you can find my comments suggestions in the notes in the text. Please pay attention to the appropriatness of the terminology used in the vaccine development field.

Kindest Regards 

Author Response

We sincerely thank the reviewers for their constructive criticisms. We have modified the manuscript accordingly. Our responses are bullet-pointed and italicized below the reviewer's comments.

Reviewer 1.

Comments and Suggestions for Authors

Dear Authors,

your manuscript need some revision to improve it and you can find my comments suggestions in the notes in the text. Please pay attention to the appropriatness of the terminology used in the vaccine development field.

Kindest Regards 

  1. Replace "deeper lung infection and severe immunopathologies with severe pneumonia, multiple organ involvement and fatal outcome.
  • We are thankful to the reviewer for pointing this out. We have replaced deeper lung infection and severe immunopathologies with severe pneumonia, multiple organ involvement, and fatal outcome in the text (Line 36)

  1. Replace "anticipated future waves of the outbreak" with recurrent epidemics.
  • The author thanks the reviewer for his/her suggestion. We have replaced anticipated future waves of the outbreak" with recurrent epidemics (Line 41).

  1. Please delete "For developing effective viral vaccines". And "The" before viral.
  • The above sentence has been deleted, and the article" The" has been added before viral in the text (Line 58).

  1. Please delete "almost always and replace it with "been identified as one of the preferred vaccine targets".
  • We have deleted "almost always" and have replaced it with "been identified as one of the preferred vaccine targets in the text (Line 59).

  1. I find the concept confused. Does the wording "delivery platform" refer to the adjuvant system, to the carrier to what? I suggest to use the wording "technical platform" as for instance attenuated viral platform do not have any delivery system than the whole virus, unless there is an adjuvant.
  • We thank the reviewer for pointing this out and have replaced the "delivery platform" with "technical platform," as suggested (Line 75).

  1. Not clear what this "COVID-19 recovered patients in future vaccine design are" sentence is about, please clarify.
  • The above sentence has been modified in the text as "Furthermore, we have discussed the immune responses elicited in recovered COVID-19 patients that can provide useful insights for vaccine design." (Line 84-85).

  1. I suggest to delete "Therefore, a multi-modal vaccine developmental approach is critical, as each strategy will target the virus uniquely owing to its different resultant immune responses and ultimately lead to protection." As it does not appear to have much sense in the context.
  • We thank the reviewer for his/her suggestions and have deleted from the text as per the reviewer's recommendation.

  1. Reviewer noted in the text "This is not a technological vaccine platform but a delivery platform as intranasal platform or needle free. I suggest to delete the all paragraph" about section 2.6: Microneedle array vaccine.
  • We are thankful to the reviewer for pointing this out. Section 2.6 has been deleted as per the reviewer's suggestion.
  1. "The purpose and aim of this chapter is not clear.I suggest to re-shape the title and the content and address the information on immune response to COVID19 infection/disease on the possibility to establish some correlates of protection that are the neutralizing antibodies.You can speculate whether NT against the Whole virus, or S protein or RBD" about section 3: Recapitulating convalescent patients' immune responses for vaccine development.
  • We are thankful to the reviewer for the insight. Now, section 3. title has been changed to "Insights from immune responses elicited in the recovered patients for vaccine development" as per reviewer suggestion (Line 322).

  1. In the conclusion section, the reviewer has heightened three words and a sentence with the notes to change the words and delete the sentence.
  • We are thankful to the reviewer for his/her suggestion. In conclusion section 5, we have changed the words and have deleted the sentence as per the reviewer's suggestion (Line 397).

Reviewer 2 Report

This manuscript presents an overview of the present status in vaccine development against the SARS-CoV-2 corona virus. It starts with an Introduction on virological aspects, and introduction to vaccines. Then a short section introduces different types of vaccines and the associated immune response. In subsequent sections details are given on various vaccines that are in development. This is followed by sections dealing with the human immune response to the virus: special attention is given to the use of plasma from patients after infection as a treatment option. A relevant section is thereafter dealing with the risks in vaccine development, especially emphasizing the need for proper efficacy and safety data when entering the market. This is followed by a conclusion section. The quality of figures is fine, and there are 84 articles in the reference list.

This manuscript provides a relevant overview of the status of coronavirus vaccines. There are a number of recent updates, especially in the field of clinical trials that can be included: the authors are referred to the section in the New York Times which has substantial and regularly updated information in the section “Coronavirus Vaccine Tracker”. There is a certain imbalance between the various sections regarding the depth of details described, but this is acceptable although too much detail affect readability.

There are a number of comments to improve the quality of the manuscript:

  • Essentially, the title is not correct. The content of the review is rather: “COVID-19 Vaccines Currently 2 Under Preclinical and Clinical Studies, and the associated antiviral human immune response”. It is advised to reconsider the title.
  • It is strongly advised to include a list of abbreviations. Also abbreviations are not always explained at first use.
  • A number of references are incomplete.
  • When referring to clinical trials it is advised to include references to clinicaltrials.gov.
  • Line 46: the change in the introduction starting with the need for a vaccine needs an introduction. It is advised to reconsider the Introduction, and if possible restructure the Introduction so that it is more logical to the reader. This includes also the position of Figure 1, which is not referenced in the text.
  • Line 85: Figure 2 is a great picture, which needs an explanation of abbreviations in the legend. In the legend “relative number” is stated, but this appears to be “absolute number”? It is advised to refer not only to the WHO but also to coronavirus vaccine tracker in the New York Times. See https://www.nytimes.com/interactive/2020/science/coronavirus-vaccine-tracker.html?name=styln-coronavirus-national&region=TOP_BANNER&block=storyline_menu_recirc&action=click&pgtype=Interactive&impression_id=f9cb5451-00a8-11eb-9ef8-c74c0d1de64b&variant=1_Show
  • Line 91, section 2: it is advised to include a description of innate immune responses in this section.
  • Same section: it is advised to prepare a summary table of the various types of vaccines detailing their structure/composition, mechanism and stage of development.
  • Line 99: Figure 2 is described as including the immune response: actually this is not the case: Figure 2 presents different types of vaccines.
  • Line 119: this statement needs a reference.
  • Line 125: reference 36 relates to clinical work and not animal models: it is advised to add data on studies in nonhuman primates. Are the sentences here about the vaccine developed in China that is these days already in use in large populations? If so, since this vaccine is criticized by the scientific community, it is advised to give this attention in the text.
  • Line 126: please add references to studies in preclinical models.
  • Line 130: Is it correct that Figure 3 presents all different types of vaccines? If not, expand this figure or create additional figures so that all vaccine types are adequately covered. Then, could this figure then be transferred to the introductory section 2 “Major COVID-19 vaccine candidates and their responses”?
  • Line 136: Since “Viral vectored vaccines” are frontrunners in clinical trials, it is advised to give more attention in the paragraph on clinical trials. Note that the vaccines developed by Johnson and Johnson, and by Astra Zeneca, are in phase 3 of clinical development. It is advised to give references to the content of these trials that has recently been published by the respective companies: note that such publications are normally kept secret.
  • Line 217: since mRNA vaccines are further in development than DNA vaccines, would it be logical to first present section 2.4 and thereafter section 2.3? Also here it is advised to give clear attention to clinical trials at the end of the section, with reference to trial description as published by the company. Note that the NIH/Moderna trial is already in phase 3.
  • Line 269: Note that Novavax just entered phase 3 clinical trials.

Author Response

RESPONSE TO REVIEWER'S COMMENTS

We sincerely thank the reviewers for their constructive criticisms. We have modified the manuscript accordingly. Our responses are bullet-pointed and italicized below the reviewer's comments.

Reviewer 2

Comments and Suggestions for Authors

This manuscript presents an overview of the present status in vaccine development against the SARS-CoV-2 corona virus. It starts with an Introduction on virological aspects, and introduction to vaccines. Then a short section introduces different types of vaccines and the associated immune response. In subsequent sections details are given on various vaccines that are in development. This is followed by sections dealing with the human immune response to the virus: special attention is given to the use of plasma from patients after infection as a treatment option. A relevant section is thereafter dealing with the risks in vaccine development, especially emphasizing the need for proper efficacy and safety data when entering the market. This is followed by a conclusion section. The quality of figures is fine, and there are 84 articles in the reference list.

This manuscript provides a relevant overview of the status of coronavirus vaccines. There are a number of recent updates, especially in the field of clinical trials that can be included: the authors are referred to the section in the New York Times which has substantial and regularly updated information in the section "Coronavirus Vaccine Tracker". There is a certain imbalance between the various sections regarding the depth of details described, but this is acceptable although too much detail affect readability.

There are a number of comments to improve the quality of the manuscript:

  1. Essentially, the title is not correct. The content of the review is rather: "COVID-19 Vaccines Currently 2 Under Preclinical and Clinical Studies, and the associated antiviral human immune response". It is advised to reconsider the title.
  • We are thankful to the reviewer for suggesting a more appropriate title for our manuscript. We have changed the title, as suggested by the reviewer (Line 2-4).

  1. It is strongly advised to include a list of abbreviations. Also abbreviations are not always explained at first use
  • We thank the reviewer for the suggestion. We have explained the abbreviation in the text and have added the abbreviation list at the end of the manuscript (Line 411).

  1. A number of references are incomplete.
  • We thank the reviewer for pointing this out and have checked all the references carefully.

  1. When referring to clinical trials it is advised to include references to clinicaltrials.gov.           

  • We have included the clinicaltrials.gov references in sections 2.1, 2.2, 2.3, 2.4, and 2.5, as suggested by the reviewer.

  1. Line 46: the change in the introduction starting with the need for a vaccine needs an introduction. It is advised to reconsider the Introduction, and if possible restructure the Introduction so that it is more logical to the reader. This includes also the position of Figure 1, which is not referenced in the text.      
  • We thank the reviewer for such valuable suggestions. We have restructured the introduction section for a more logical flow for the readers (Line 30-54). Figure 1 has been referred in the text (Line 58).

  1. Line 85: Figure 2 is a great picture, which needs an explanation of abbreviations in the legend. In the legend "relative number" is stated, but this appears to be "absolute number"? It is advised to refer not only to the WHO but also to coronavirus vaccine tracker in the New York Times. See https://www.nytimes.com/interactive/2020/science/coronavirus-vaccine-tracker.html?name=styln-coronavirus-national&region=TOP_BANNER&block=storyline_menu_recirc&action=click&pgtype=Interactive&impression_id=f9cb5451-00a8-11eb-9ef8-c74c0d1de64b&variant=1_Show

  • We thank the reviewer for appreciating our Figure 2. We appreciate the reviewer for pointing out our relative number mistake, which has now been changed to "number" (Line 88). We have also referred coronavirus tracker and WHO, as suggested by the reviewer (Line 91), and have also explained the abbreviation in the legend (Line 91-92).

  1. Line 91, section 2: it is advised to include a description of innate immune responses in this section.
  • The authors thank the reviewer for his/her advice. In section 2, a brief description of the innate immune response has been included (Line 106-109).

  1. Same section: it is advised to prepare a summary table of the various types of vaccines detailing their structure/composition, mechanism and stage of development.  
  • We thank the reviewer for his/her suggestions. A table listing various types of vaccines, their composition, developer, and development stage has been added in section 2 (Line 119-120). A brief mechanism is also added in the text (Line 109-115).

  1. Line 99: Figure 2 is described as including the immune response: actually this is not the case: Figure 2 presents different types of vaccines.
  • The authors thank the reviewer for pointing this out. It appears the reviewer is talking about Figure 3, not Figure 2. The figure 3 description (Line 103-104) and legend have been modified as suggested by the reviewer (Line 119).

  1. Line 119: this statement needs a reference.
  • The authors thank the reviewer for pointing this out. The relevant reference has been added as per the reviewer's suggestion (Line 142).

  1. Line 125: reference 36 relates to clinical work and not animal models: it is advised to add data on studies in nonhuman primates. Are the sentences here about the vaccine developed in China that is these days already in use in large populations? If so, since this vaccine is criticized by the scientific community, it is advised to give this attention in the text.
  • We thank the reviewer for correcting us. We have corrected the sentence and have added the correct reference (Line 148). This is regarding BBIBP-CorV vaccine developed by Sinopharma, China. We are not aware of this vaccine's large-scale uses and related controversy; thus, we restrain ourselves from commenting on it.

  1. Line 126: please add references to studies in preclinical models.
  • The authors thank the reviewer for pointing this out. The relevant reference has been added as per the reviewer's suggestion (Line 152).

  1. Line 130: Is it correct that Figure 3 presents all different types of vaccines? If not, expand this figure or create additional figures so that all vaccine types are adequately covered. Then, could this figure then be transferred to the introductory section 2 "Major COVID-19 vaccine candidates and their responses"?
  • We thank the reviewer for his/her concern regarding Figure 3. As per our understanding, Figure 3 is correct, and it represents all different types of vaccine platforms.

  1. Line 136: Since "Viral vectored vaccines" are frontrunners in clinical trials, it is advised to give more attention in the paragraph on clinical trials. Note that the vaccines developed by Johnson and Johnson, and by Astra Zeneca, are in phase 3 of clinical development. It is advised to give references to the content of these trials that has recently been published by the respective companies: note that such publications are normally kept secret.
  • We thank the reviewer for the advice. Yes, the reviewer correctly pointed out that the "Viral vectored vaccines" are frontrunners in clinical trials. We have given more attention to Johnson and Johnson and Astra Zeneca's clinical trials and have added their details and relevant references in section 2.2 (Line 156-193).

  1. Line 217: since mRNA vaccines are further in development than DNA vaccines, would it be logical to first present section 2.4 and thereafter section 2.3? Also here it is advised to give clear attention to clinical trials at the end of the section, with reference to trial description as published by the company. Note that the NIH/Moderna trial is already in phase 3.
  • The authors thank the reviewer for the suggestions. We have interchange the sequence of sections 2.3 and 2.4 as per the reviewer's advice. We have described the clinical trials in greater detail and have added relevant references in section 2.3 (Line 224-230).  We also made a note about the NIH/Moderna trial entering phase 3 (Line 225).

  1. Line 269: Note that Novavax just entered phase 3 clinical trials.
  • The authors thank the reviewer for pointing the clinical phase of Novavax. It has been corrected to clinical phase 3 (Line 294).

Round 2

Reviewer 1 Report

Dear Authors,

I have received your cover letter and the revised paper.

Thanks for accepting all my suggestions.

The manuscript can be published in the revised form.

Kindest Regards